# Novel Technology for Enamel Remineralization in Artificially Induced White Spot Lesions: In Vitro Study

**Lavinia Luminita Voina Cosma** [1], **Marioara Moldovan** [2], **Alexandrina Muntean** [1], **Cristian Doru Olteanu** [3], **Radu Chifor** [4,*] and **Mindra Eugenia Badea** [4]

[1] Department Pedodontics, University of Medicine and Pharmacy "Iuliu Hatieganu", 400083 Cluj-Napoca, Romania
[2] "Raluca Ripan" Institute for Research in Chemistry, University "Babes-Bolyai", 400294 Cluj-Napoca, Romania
[3] Department Orthodontics, University of Medicine and Pharmacy "Iuliu Hatieganu", 400083 Cluj-Napoca, Romania
[4] Department Prevention in Dentistry, University of Medicine and Pharmacy "Iuliu Hatieganu", 400083 Cluj-Napoca, Romania
* Correspondence: chifor.radu@umfcluj.ro; Tel.: +40-742195229

**Abstract:** The enamel white spot lesion is a common complication of orthodontic treatment with a high prevalence. This research aims to create an artificially induced white spot lesion, evaluate three different commercial products in terms of visual appeal, mineral reestablishment, and roughness, and determine which material can recover the initial structure. We created an artificially induced white spot lesion in extracted teeth. The materials used in the study were peptide p11-4 (CurodontTM Repair, Credentis AG), bioactive glass toothpaste (Biomin F, BioMin Technologies Limited), and local fluoridation (Tiefenfluorid, Humanchemie) in conjunction with low-level laser therapy (LLLT). To objectively assess the surface, the roughness, mineral content, and esthetic were measured. The roughness increased with a median difference of $-0.233$ μm in the bioactive glass group; the color parameter delta L decreased dramatically with a median difference of 5.9–6.7; and the cervical third increased the Ca-P mineral content above the starting stage. Each material contributed significantly to enamel consolidation, with peptide therapy providing the most encouraging results.

**Keywords:** bioactive glass; fluoride; laser; peptide; remineralization; white spot lesion

## 1. Introduction

The literature defines a white spot lesion (wsl) as an initial, non-cavitated, and active caries in the tooth's enamel. This surface is recognized and macroscopically altered in a white/opaque formation with an uninterrupted external layer [1,2]. Enamel initial demineralization is the most frequent complication during orthodontic treatment [3,4]. The plaque accumulation of bacteria around the bracket is the etiology of this condition, which encourages an acidic attack on the tooth's outer shell, disintegrating the enamel minerals [4–7]. According to Chapman et al., the ratio of this disease in the upper teeth divides as follows: 34% for the lateral incisor, 31% for the canine, 28% for the first premolars, and only 17% for the central incisors [4,5]. This lesion's consistency appears on the buccal maxillary surface, the middle third of the tooth around the bracket, and the cervical third [8]. The accuracy of white spot lesions for patients undergoing fixed orthodontic treatment ranges between 2–97 percent, and it has been highlighted four weeks after the application [2,7,9]. We chose this theme because treating this pathology remains challenging due to the complexity of the enamel structure and individual customs. Numerous attempts in the literature to find a remedy have resulted in two distinct paths: remineralization or camouflage [4,7,9]. The camouflage technique employs resin infiltration, whereas the second method, which is the most debated and is currently being researched, refers to mineral structure recapturing. The remineralization approach offers a variety of options,

ranging from the commonly used fluoride and casein phosphopeptide-amorphous calcium phosphate products in various forms to the newly added substances in adhesion, pastes, or solutions, such as bioactive glass, nanohydroxyapatite, peptide p11-4, and cold plasma [10–15]. The evolution of research in dental prophylaxis is ongoing to identify the best product for caries prevention and the remineralization of the white spot lesion. Fluoride is an essential agent in caries prevention because it inhibits tooth demineralization; therefore, many different types of fluoride with different concentrations, releasing systems, and enrichments with other substances have been developed.

Tiefenfluorid is a material that provides deep-penetration fluoridation by precipitating calcium fluoride in the funnels of the loosened enamel (approx. 7 m) of hard tooth tissue. It generates spontaneously in a precipitation reaction, in addition to magnesium fluoride and silica gel, after applying the solutions [16,17]. In the literature, few articles have tested this novel therapeutic approach. Laser irradiation has been used for a while in studies for caries prevention. It has been demonstrated that it produces a significant decrease in dissolution at the surface of the enamel, as well as fusion and recrystallization of hydroxyapatite crystals that are more resistant to acidic solutions [18,19]. The combination of fluoride with low-level laser therapy has been less studied. BioMin F technology, which contains fluoro calcium phosphosilicate bioactive glass, was recently introduced. Any substance that can form a hydroxyl-carbonated apatite layer within a biological system is considered a bioactive material. The bioactive glass material interacts with cells and tissues and starts a layer inside the saliva. The general similarity in the chemical constituents of enamel and bone material has increased interest in dentistry [11,20–22]. Another material recently gaining popularity is the peptide p11-4, a class of peptides that goes through a hierarchical order and a predetermined process of scaffold assembly and formation. Curodont Repair is a guided enamel regeneration product that uses the p11-4-based Curolox technology. By binding phosphate and calcium ions from saliva, this regeneration process restores the original composition of the enamel by inducing de novo hydroxyapatite crystal growth [14,23].

The advancement and development of technologies and materials are currently putting a strain on society. Testing the products is required to assess the materials and their impact and to narrow the knowledge gap. Experiments are classified into three types: in vitro, in vivo, and in silico, each playing an essential role in research. In vitro research involves studying tissues, human cells, animal cells, or bacteria outside a living organism. The investigations are well controlled, making them suitable for studies requiring a particular target; however, they can only partially replicate natural functioning; it is difficult to predict what would happen within an organism, and the results obtained may differ from time to time. In vivo studies are carried out within a living organism, including animal testing and clinical trials on human applicants. The advantage of these studies is that they can measure the effects of mixtures and are standardized. In contrast, animal experiments require significant resources, and only a few species represent a large ecosystem. An in silico experiment is conducted using computer software or a computer simulation. It is the most recent of the three research methods and has significantly contributed to biomedicine research and clinical trials. They do not require synthesis or preparation; toxicity can be determined before the materials enter production. As a disadvantage, it is critical to check the data quality because it can distort the results; they do not prove an experimental result and require validation to demonstrate predictability [24,25]. We chose to conduct the study in vitro because it gave us greater control over the environment in which we worked. We wanted to verify the biomechanical efficiency of these materials through roughness and the restoration of the composition of the dental hard tissue through X-ray, neither of which can be performed in vivo or in silico.

The novelty of this work originated from the testing of fluoride with low-level laser therapy, as well as the comparison of these three products in terms of esthetics, the remineralization effect by mineral content regain, mechanical testing by roughness, and the plaque accumulation susceptibility.

The objectives of this paper are to create an artificially induced white spot lesion, to treat this lesion with three novel technology materials, and to compare it in relation to esthetics and physical and remineralization terms and to determine which material provides the best outcome for recovering the original structure.

## 2. Materials and Methods

The study was conducted at the "Raluca Ripan" Institute for Research in Chemistry at the University "Babes-Bolyai" in Cluj-Napoca, Romania, and the Department of Prevention in Dentistry at the University of Medicine and Pharmacy "Iuliu Hatieganu" in Cluj-Napoca, Romania. Fifty-four randomly selected teeth (19 incisors, 6 canines, 4 premolars, and 25 molars) from periodontal disease patients were used in the study. The inclusion criteria for the tooth selections were no macroscopical cracks, hypoplasia, or caries lesions on the tooth's buccal surface. The exclusion criteria were visible cracks, fractures, or decay on the enamel. The probes were scaled and polished before being placed in a silicon cup filled with acrylic resin (Duracryl Plus Spofa Dental) until the buccal outer layer of the teeth was sectioned with a microtome machine, yielding slices of enamel ranging in thickness from 1–3 mm. The buccal outer layer of the enamel, which was kept intact after vertical sectioning, was the tested surface of the tooth and had a size of approximately 8 mm × 10 mm. We wanted to replicate the conditions found in the oral cavity using artificial saliva; the template was $Na_2HPO_4$ 0.426 g, $NaHCO_3$ 1.68 g, $CaCl_2$ 0.147 g, $H_2O$ 800 mL, and HCl-1 M 2.5 mL, and the pieces were kept at room temperature (25 °C) throughout the experiment. The research strategy, divided into three stages, is summarized in Scheme 1, as are the chemical compositions of the materials used and the manufacturer information. The primary step coincided with the teeth preparation. The second step met the intermediate phase, where a 37 percent ($H_3PO_4$) acid etching solution was prepared to assess white spot lesions, immersing the teeth for 4 min. In the last step, the teeth were divided into three groups to test different therapeutic approaches. The teeth were split into groups using the random.org website. The first option was to combine local fluoridation (Tiefenfluorid Humanchemie) with low-level laser therapy (group F+LLLT). The teeth were dried with a cotton roll before applying Tiefenfluorid Tochierlosung with an applicator. After the solution was assimilated, Tiefenfluorid Nachtouchierlosung was used and irradiated with the laser Sirolaser Blue Sirona at a distance of 4 mm, with a wavelength of 660 nm, a power of 100 mW, and a time of irradiation of 60 s. Biomin F, a toothpaste enriched with a bioactive glass (BAG), was the second treatment option (group BAG). A cotton roll was used to dry the teeth, and a 1 cm toothpaste was applied twice daily for 2 min until the fourth day. Peptide p11-4 (CurodontTM Repair) was used in the final alternative therapy (group p11-4). The teeth were dried with cotton rolls, disinfected with sodium hypochlorite 2% (Chloraxid 2% Cerkamed) for 20 s, etched with acid orthophosphoric 35% for 20 s, washed, and dried. Finally, Curodont repair was utilized and allowed to diffuse for 5 min.

The surface measurements were made in all stages (initial—before creating wsl, intermediate—after completing the wsl, and final—after the treatment application), evaluating the teeth's surface roughness, color, and enamel mineral stability. The roughness was measured optically on the Alicona Infinite Focus microscope (Alicona Imaging GmbH, Graz, Austria), which provides 3D surface quantification by integrating the absolute value of roughness (Ra). This machine precisely measures tools for tolerances in the μm and sub-μm ranges. The surface was scanned over a 4 $mm^2$ area in the middle third of the tooth. A 2 mm line was drawn in the area with no scanning gaps, and a cut-off (λ c) of 250,000 μm was used to process the measurement in accordance with DIN EN ISO 4288. A high Ra value influences plaque accumulation, which could favor dental caries over time.

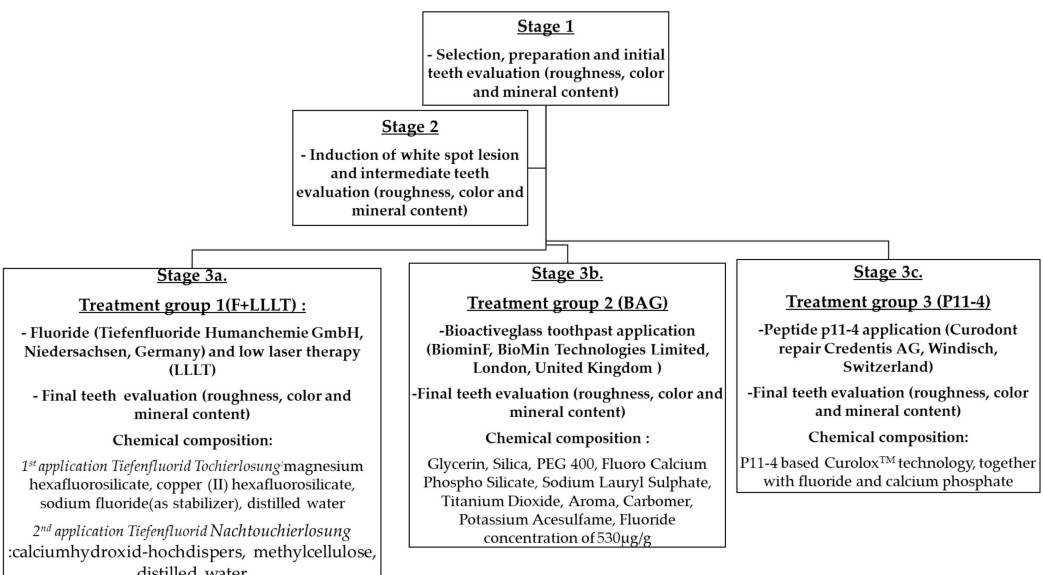

**Scheme 1.** The research's strategy.

The color parameter was measured using the Vita Easy Shade spectrophotometer (Vita Zahnfabrik, Bad Säckingen, Germany). This digital device was designed to identify the shade of natural teeth and ceramic restorations precisely, quickly, and reliably. The accuracy is very high, 93.75%, and the measurement's reliability is based on LED technology, which is unlikely to be affected by environmental conditions. The spectrophotometer calculates the CIELAB (Commission Internationale de L'Eclairage L*a*b) color notation system. Before each evaluation, the probe tip was calibrated on the calibration port built into the machine. The teeth were measured by holding the device tip 90 degrees to the surface in the middle third of the teeth. All samples were analyzed using measurement methods of lightness (parameter L) and color (parameters a for red/green and b for yellow/blue). The measurements were formalized using the same background, operator, and lighting conditions. The Fischerscope X-ray fluorescence analysis (Helmut Fischer GmbH, Sindelfingen, Germany) was used to determine the enamel's mineral content. The software converts the data from the measured X-ray spectra into parameters for layer thickness measurement and material analysis. The technique is based on the fact that when atoms are excited by primary X-rays, they discharge power in the form of element-specific fluorescence radiation. The spectrum of the energy radiated reveals information about the sample's composition. The detector has a high energy resolution and, therefore, can provide precise, measured data in a short amount of time. For each treatment group, we examined the report between the elements Ca (Calcium) and P (Phosphorus) and Ca/F (Fluoride) in four regions: the middle third of the incisal, cervical, mesial, and distal zone. The results are interpreted by analyzing whether the initial report is re-established. All the data from the study were analyzed using IBM SPSS Statistics 25 and illustrated using Microsoft Office Excel/Word 2013. Quantitative variables were tested for normal distribution using the Shapiro–Wilk test and were written as averages with standard deviations or medians with interquartile ranges. Quantitative independent variables with non-parametric distribution were tested using the Mann–Whitney U/Kruskal–Wallis H tests. Quantitative independent variables with normal distribution were tested using the Student/One-Way ANOVA/Welch ANOVA tests. Post-hoc analysis was made using the Tukey HSD/Games–Howell/Dunn–Bonferroni tests. Quantitative variables with repeated measures and non-parametric distribution were tested using related samples Wilcoxon signed-rank tests. Quantitative variables with repeated measures and normal distribution were tested using paired-samples *t*-tests. In each treatment group, the comparison was made according to the distribution of the measured intervals.

## 3. Results

### 3.1. Surface Roughness Measurement Ra Parameter in Each Treatment Group

The evolution of the Ra parameter in each treatment group is shown in Figure 1. According to the Wilcoxon tests, the Ra parameter did not change significantly in evolution in the F+LLLT and p11-4 peptide groups (median difference of −0.038 and 0.031 µm), and the observed difference in Ra was not statistically significant. The Ra parameter in the BAG group increased significantly in evolution ($p = 0.039$), with a significant difference (median = −0.233 µm, IQR = −0.473–0.140 µm).

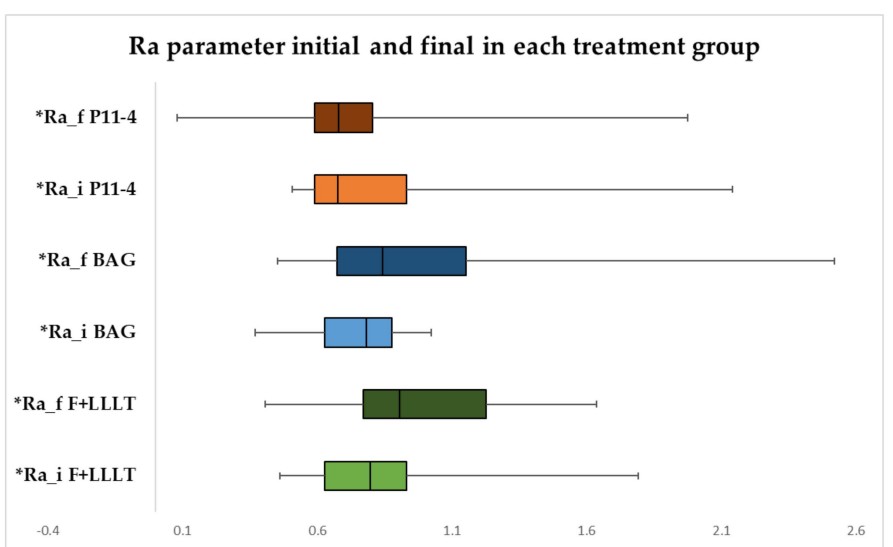

**Figure 1.** Boxplot representation of Ra parameter initial and final in each treatment group. * Ra_i−parameter Ra initial; Ra_f−parameter Ra final.

### 3.2. Color Measurement of "L" Parameter in Each Treatment Group

The progression of the L parameter is represented in Figure 2, and according to the paired-samples *t*-tests/Wilcoxon tests, the F+LLLT and BAG groups reduced dramatically in advancement, with a significant difference in comparison to the p11-4 peptide group, where there was no change in evolution and statistical significance.

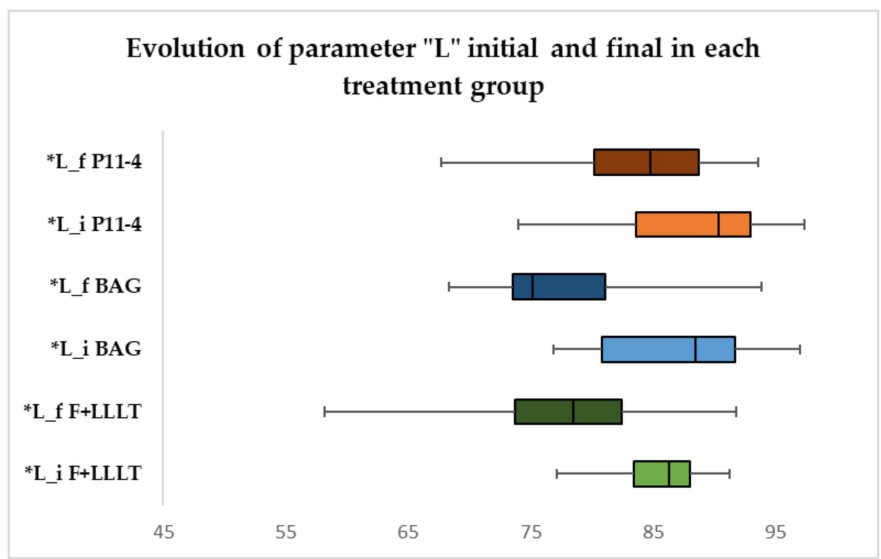

**Figure 2.** Boxplot representation of the evolution of "L" parameter initial and final in each treatment group. * L_i−color parameter "L" initial; L_f−color parameter "L" final.

### 3.3. Colour Measurement of "a" Parameter in Each Treatment Group

The evolution of the a parameter is shown in Figure 3; it rose exponentially in progression ($p < 0.001$) in all three treatments, with a significant difference.

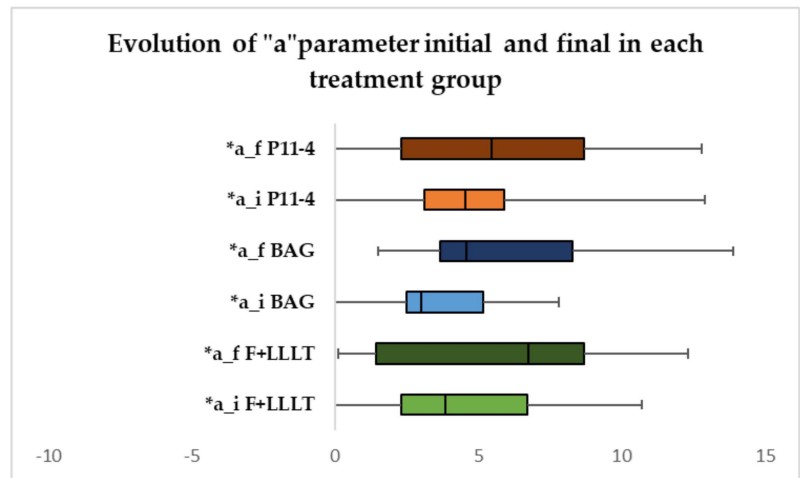

**Figure 3.** Boxplot representation of the evolution of "a" parameter initial and final in each treatment group. * a_i—color parameter "a" initial; a_f—color parameter "a" final.

### 3.4. Colour Measurement of "b" Parameter in Each Treatment Group

The boxplot from Figure 4 exposes the evolution of the b parameter in each treatment group. The paired-samples *t*-tests show that the parameter changed radically in all treatment groups, with a significant difference.

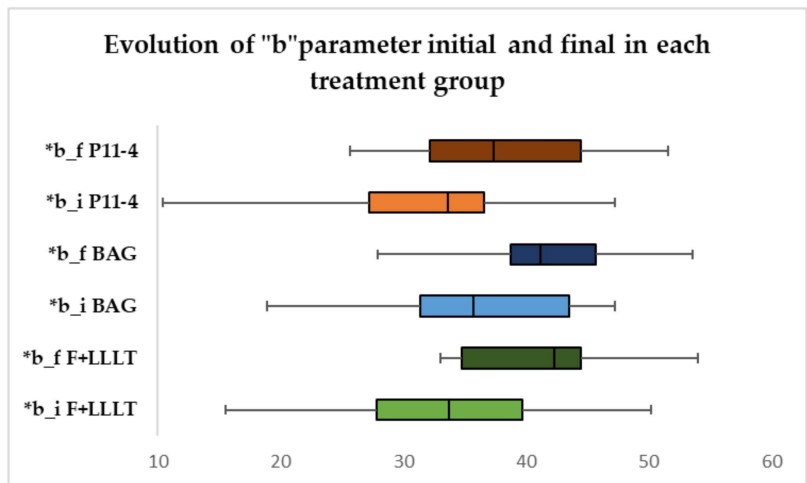

**Figure 4.** Boxplot representation of the evolution of "b" parameter initial and final in each treatment group. * b_i—color parameter "b" initial; b_f—color parameter "b" final.

### 3.5. Evolution of the Incisal Third Ca/P and Ca/F Ratio in Each Treatment Group

The data from Figure 5 show the evolution of the incisal third Ca/P ratio in each treatment group. With regard to the Wilcoxon tests, the results show that in all the groups the Ca/P ratio did not change significantly in evolution. The observed difference was not statistically significant. The incisal third Ca/F ratio variation is represented in Figure 6, and according to the Wilcoxon tests, the treatment groups were not altered in progression. The observed difference was not statistically significant.

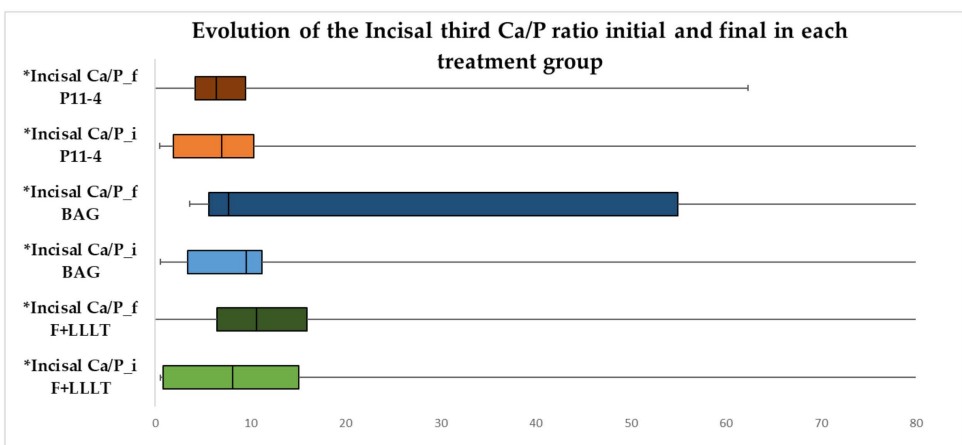

**Figure 5.** Boxplot representation of the evolution of the incisal third Ca/P ratio initial and final in each treatment group. * Incisal Ca/P_i—incisal third Ca/P ratio initial; incisal Ca/P_f—incisal third Ca/P ratio final.

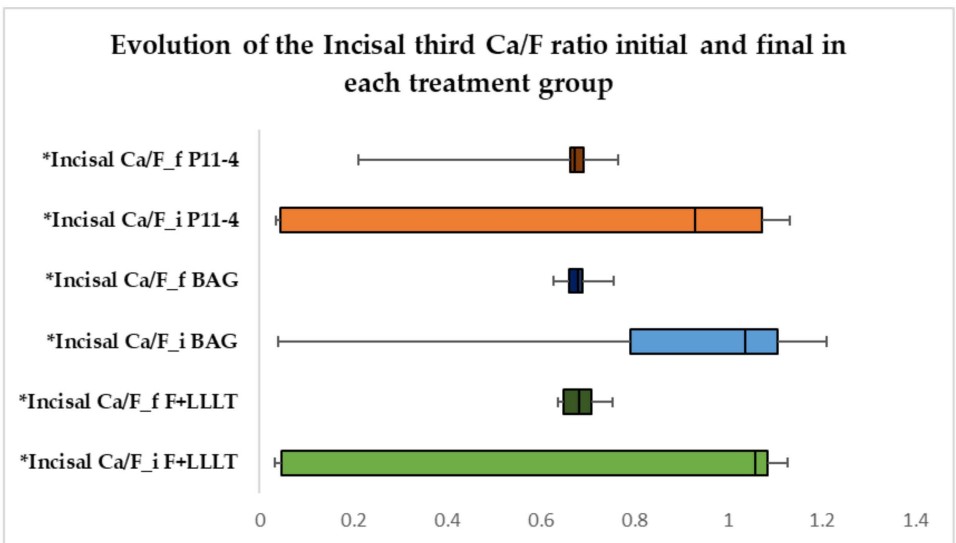

**Figure 6.** Boxplot representation of the evolution of the incisal third Ca/F ratio initial and final in each treatment group. * Incisal Ca/F_i—incisal third Ca/F ratio initial; incisal Ca/F_f—incisal third Ca/F ratio final.

*3.6. Evolution of Cervical Ca/P and Ca/F Ratio in Each Treatment Group*

Figure 7 records show the effects of the cervical Ca/P proportion in each treatment. According to the Wilcoxon tests, the Ca/P ratio in the p11-4 peptide group increased significantly in evolution ($p = 0.022$), with a significant difference (median = −2.912, IQR = −4.157–1.250) in reference to the other groups where the ratio did not change significantly in evolution, and the observed difference was not statistically significant. The development of the cervical Ca/F ratio for each treated group can be seen in Figure 8. According to the Wilcoxon tests in all the groups, the Ca/F balance did not significantly change in evolution and the observed difference was not statistically significant.

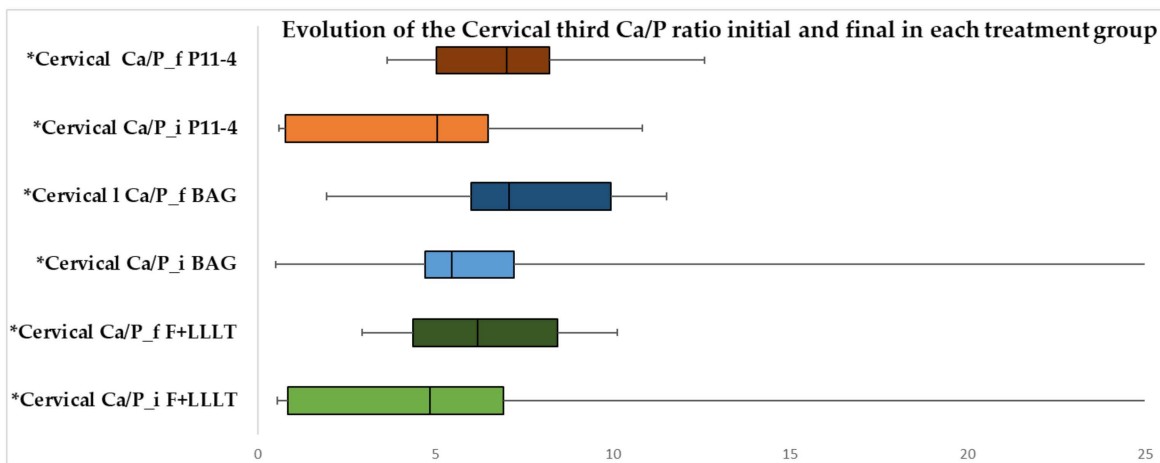

**Figure 7.** Boxplot representation of the evolution of Ca/P ratio in the cervical third initial and final in each treatment group. * Cervical Ca/P_i−cervical third Ca/P ratio initial; cervical Ca/P_f−cervical third Ca/P ratio final.

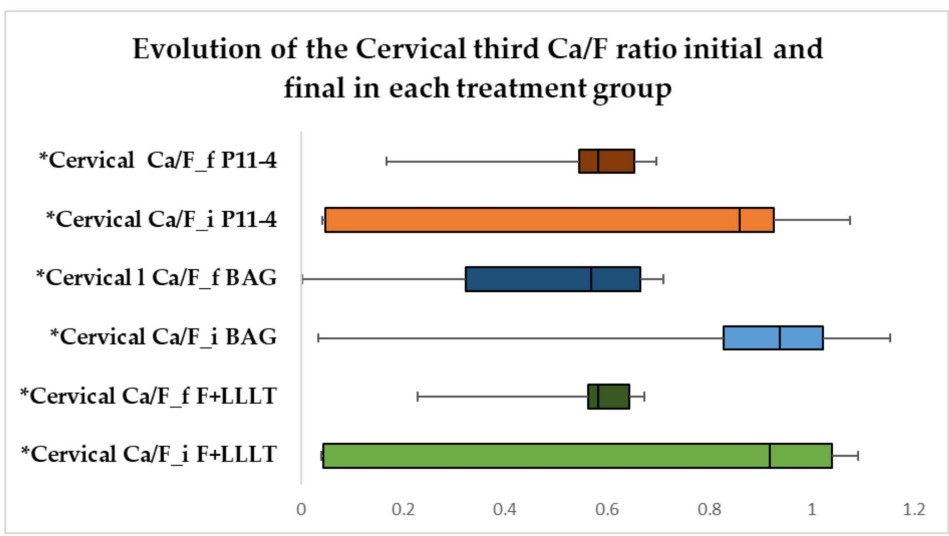

**Figure 8.** Boxplot representation of the evolution of Ca/F ratio in the cervical third initial and final in each treatment group. * Cervical Ca/F_i−cervical third Ca/F ratio initial; cervical Ca/F_f−cervical third Ca/F ratio final.

*3.7. Evolution of Mesial Ca/P and Ca/F Ratio in Each Treatment Group*

The data in Figure 9 show the variation of the mesial Ca/P balance in each treatment group. The results indicate that the Ca/P ratio did not change considerably in all the groups over time, and the identified difference is insignificant. The data from Figure 10 show the existence of the mesial Ca/F ratio in each treatment group. The findings suggest that the mesial Ca/F ratio in the BAG group dropped dramatically in evolution ($p = 0.016$), with a significant difference (median = 0.422, IQR = 0.19–0.497) compared to the other groups, where the ratio did not change significantly in advancement, and the observed difference is not statistically significant.

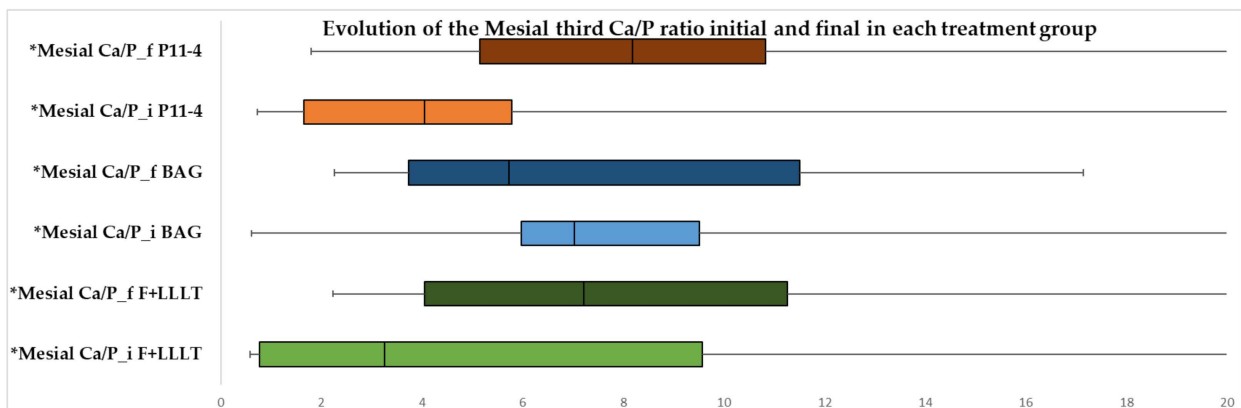

**Figure 9.** Boxplot representation of the evolution of Ca/P ratio in the mesial third initial and final in each treatment group. * Mesial Ca/P_i—mesial third Ca/P ratio initial; mesial Ca/P_f—mesial third Ca/P ratio final.

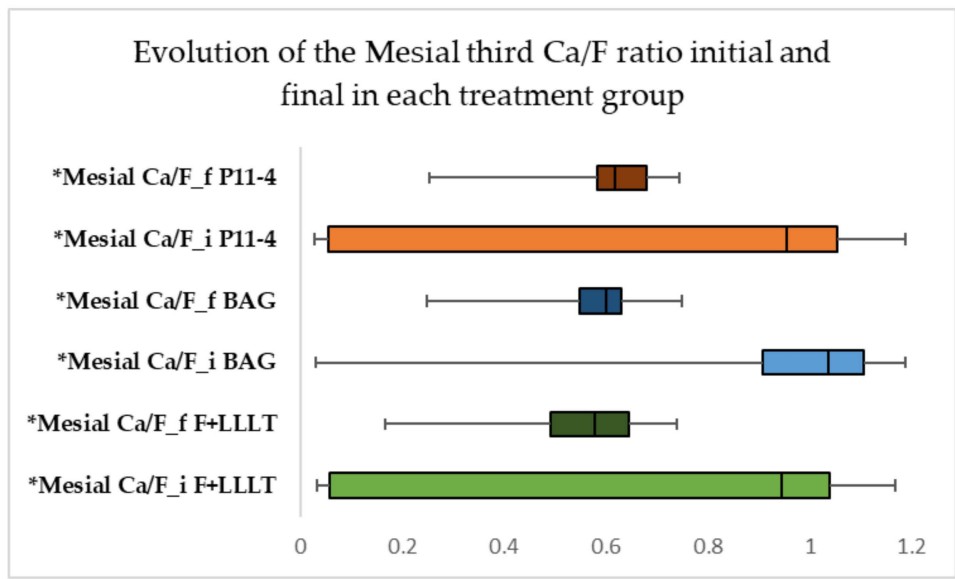

**Figure 10.** Boxplot representation of the evolution of Ca/F ratio in the mesial third initial and final in each treatment group. * Mesial Ca/F_i—mesial third Ca/F ratio initial; mesial Ca/F_f—mesial third Ca/F ratio final.

*3.8. Evolution of Distal Ca/P and Ca/F Ratio in Each Treatment Group*

The data in Figure 11 show the evolution of the distal Ca/P ratio in each treatment group. According to the Wilcoxon tests, the distal Ca/P ratio in all three treatments did not alter in evolution, and the reported difference was not statistically significant. The data from Figure 12 show the development of the distal Ca/F ratio in each treatment group; the distal Ca/F ratio in the BAG group decreased significantly in evolution ($p = 0.048$), with a significant difference (median = 0.374, IQR = 0.053–0.515). The ratio did not change significantly over time compared to the other groups, and the reported difference was not statistically considerable.

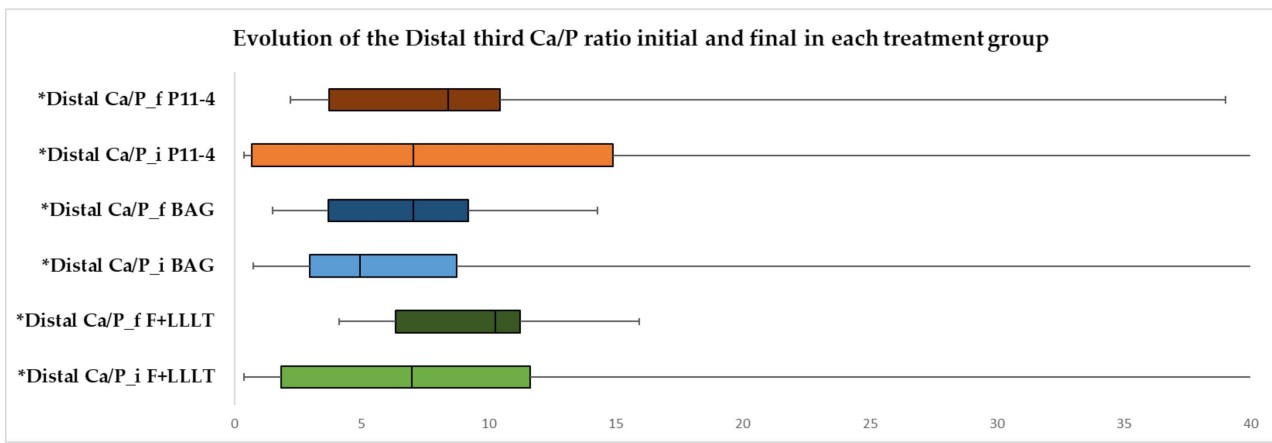

**Figure 11.** Boxplot representation of the evolution of Ca/P ratio in the distal third initial and final in each treatment group. * Distal Ca/P_i—distal third Ca/P ratio initial; distal Ca/P_f—distal third Ca/P ratio final.

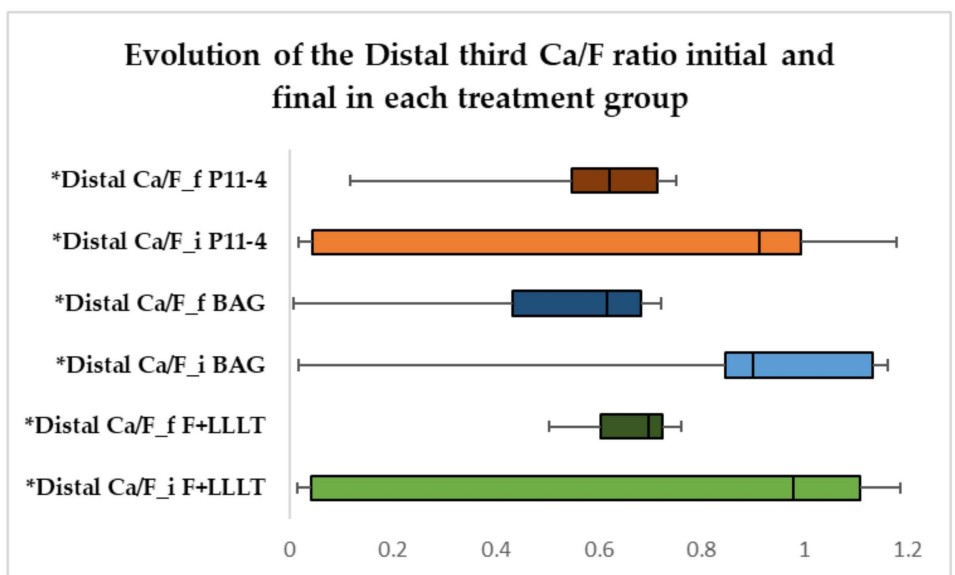

**Figure 12.** Boxplot representation of the evolution of Ca/F ratio in the distal third initial and final in each treatment group. * Distal Ca/F_i—distal third Ca/F ratio initial; distal Ca/F_f—distal third Ca/F ratio final.

## 4. Discussion

An imbalance of the demineralization and remineralization process provokes the mechanism involved in the white spot lesion. The alteration depends on the variations of the oral environment with regard to their duration and intensity. Demineralization is reversible as long as the conditions necessary for remineralization are met and the organic matrix is intact. The process's dynamic is based on the phenomenon of the dissolution of the apatite crystal and precipitation of salts in the fluid of the bacterial biofilm [26–28]. Remineralization does not reconstruct the initial enamel prism architecture, but it does form a thick layer of calcium phosphate and fluoride. This dense layer becomes more durable to further demineralization than regular enamel [29].

The study's objectives were met, and each treatment significantly contributed, more or less, to restoring the teeth's surface. The novelty of this research was the comparison of the efficacy of three different commercial products, the use of fluoride combined with low laser therapy, and the comparison of treatment achievements not only esthetically but also structurally. The microstructure of the enamel is fully accountable for its natural

roughness. Enamel etching and material use may make restoring the enamel's original condition challenging. Dental materials can potentially roughen the teeth's outer layer, influencing plaque accumulation and discoloration [30]. Our findings showed no difference between the group with the fluoride and low-level laser therapy and the peptide p11-4 before and at the end of the treatment. This discovery could imply that the initial roughness of the enamel was maintained, which is consistent with the findings of Sindhura et al. and Magalhaes et al., who demonstrated the recovery of the tooth's exterior layer in just seven days after treatment [31,32]. The bioactive glass group showed increased roughness after treatment, indicating a greater preference for enhancing bacterial deposition and raising the risk of demineralization. The last statement contradicts the study of Farooq et al., in which commercial fluorided bioactive glass toothpaste reduced roughness values, with the maximum decrease obtained using a combination of theobromine and fluoride bioactive glass toothpaste [33]. Other published studies also confirmed roughness reduction after using bioactive glass [34,35].

The color of a natural tooth is defined by the projection of incident light [36]. This feature has been utilized to develop diagnostic techniques for caries lesions relying on enamel fluorescence. Color analyses on enamel decay are rarely performed, even though they appear in the initial phases as increased whiteness that raises concerns and may need restorative approaches [37,38]. A spectrophotometer was used to properly assess the modification in color parameter (L*a*b) for the conventional quantitative assessment of coloration. The reliability and validity of the measurements have contributed to its popularity [38,39]. The L parameter represents the color's lightness [40]. Many studies have shown that creating the wsl raises the whiteness of the tooth by affecting the absorbing light in that region, implying the treatments' influence on the teeth's brightness [41]. In comparison to the peptide group, our findings show that the groups F-LLLT and BAG had a significant reduction in tooth color. Clinically, this affirmation shows that the color for the last two groups darkened in contrast to the baseline color. This result was similar to the findings of Mohamed et al. but contrary to the research results of Yetkiner et al. They investigated the color stability for fluoride treatments and accomplished a color improvement near the baseline level. Iwami et al. demonstrated that the L and components are related to caries activity; therefore, a low value may indicate a high bacterial activity that leads to enamel decay. This observation may imply that the enamel remineralization in groups F-LLT and BAG was incomplete due to a lack of deep material infiltration in the created white spot lesion and the persistence of discoloration [42–44]. The red-green chromaticity increased in all groups, indicating a transition to the red component, which is consistent with the findings of Mohamed et al. The yellow-blue chromaticity also rose in all groups except the peptide group, resulting in a much more yellow component, which contradicts the outcomes of Mohamed et al.; Polo et al. explored natural tooth color prediction in Caucasians from Spain [45]. The study's conclusion revealed that teeth appear darker, yellow, and reddish with age. This is similar to our observations because we used extracted teeth from Caucasians in Romania with periodontal disease. Consequently, this remark may indicate that all treatment groups display asymmetrical coloristic improvement.

The presence of a white spot lesion induces enamel mineral loss, implying degradation of the dental tissue, which leads to a change in the structure, sensitivity, and esthetics [46]. The mineral constituent of the tooth's outer layer is substituted calcium hydroxyapatite (HAP). Numerous element substitutions can replace the concentration of ions missing from HAP, such as calcium replacing magnesium, carbonate replacing phosphate, and hydroxyl replacing fluoride [47–49]. These changes can affect HAP's actions, particularly its dissolution rate at low pH. Calcium deficiency and carbonate-rich areas are likely exposed to acid demineralization but replacing hydroxide with fluoride increases protection against demineralization [50]. When the caries threat is prevented, it is acknowledged that white spot lesion seems to regress with the appearance of remineralizing agents [51–53]. The remineralization process is not mineral precipitation onto the tooth structure but

rather a crystal restoration in the lesion's subsurface [54]. We evaluated three different remineralization therapies in different regions of the buccal surface of the tooth in this study (middle incisal, cervical, mesial, and distal third). Compared to the other regions, the mineral gain of Ca and F in the BAG group decreased significantly in the mesial and distal middle thirds. These results have many possible interpretations. The mineral loss and regain may vary between thirds, the fluoride intake from the BAG group may diffuse slowly in comparison with the other buccal areas, and the use of the commercial products Tiefenfluorid and Curodont repair may have a better intake of the minerals. Calcium is the favored element for dissolution during the first four hours of demineralization [55]. When there is a calcium deficiency, there is selective absorption of calcium and a tendency to return to a Ca/P apatite, as demonstrated by the precipitation of calcium phosphates on hydroxyapatite crystals [56,57]. In our study, all the groups experienced a significant recovery in Ca/P, but only the peptide group gained a higher value after the treatment application in the cervical third. This fact is important because the cervical third of the tooth is known to be the least mineralized zone, with increased porosity and a higher risk of caries [58,59]. The Biomin F treatment was the least effective in returning the tooth to its original state; we encountered roughness, color, and mineral uptake issues. Tiefenfluorid and low-level laser therapy was the second effective remedy; the difficulties were noticed only with the esthetics of the enamel reformation. Finally, the peptide p11-4 Curodont repair commercial product delivered the best performance, improving all measurements and ensuring the recovery of the lost surface.

The limitations of these studies include the small number of teeth used, the lack of oral cavity conditions, the demineralization process being more aggressive than a typical acid attack in the oral cavity, and the need for additional analyses to confirm crystal reintegration. Another limitation of our study was the experiment type, in vitro, because it replicates only a portion of the natural functioning of an organism, in our case the oral cavity, and the results obtained may differ from those obtained in vivo or in silico.

## 5. Conclusions

P11-4 peptide Curodont Curolox technology had the best outcome in terms of improving white spot lesions compared to the other tested materials. However, more research is needed to confirm the research approach. We believe that developing new technologies and materials will help us understand the remineralization process of the enamel white spot lesion and that this lesion will be history in the future.

**Author Contributions:** Conceptualization, L.L.V.C. and M.E.B.; methodology, L.L.V.C., M.E.B., M.M. and R.C.; software, C.D.O.; validation, M.E.B., M.M., C.D.O. and A.M.; formal analysis, C.D.O. and A.M.; investigation, L.L.V.C. and R.C.; resources, C.D.O. and A.M.; data curation, A.M.; writing—original draft preparation, L.L.V.C.; writing—review and editing, R.C. and M.E.B.; visualization, M.M., C.D.O. and A.M.; supervision, M.E.B. and M.M. All authors have read and agreed to the published version of the manuscript.

**Funding:** This research received no external funding.

**Institutional Review Board Statement:** The study was conducted in accordance with the Declaration of Helsinki and approved by the Ethics Committee of the University of Medicine and Pharmacy "Iuliu Hatieganu" Cluj-Napoca, Romania (protocol code 228 and date of approval 25 July 2022).

**Informed Consent Statement:** Not applicable.

**Data Availability Statement:** Not applicable.

**Acknowledgments:** This research takes a part from the Ph.D. thesis entitled "Prevention and management of white spot lesion during or after orthodontic treatment", from the University of Medicine and Pharmacy "Iuliu Hatieganu" Cluj-Napoca, Romania.

**Conflicts of Interest:** The authors declare no conflict of interest.

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
