# Peer review of "Novel Technology for Enamel Remineralization in Artificially Induced White Spot Lesions: In Vitro Study"

_coatings, doi:10.3390/coatings12091285_

Round 1
Reviewer 1 Report
The paper is very interesting and will be of interest to the readers of Coatings. The Introduction provides sufficient information on the field and main objectives of the paper. The study design is appropriate and obtained results have been properly discussed.
However, the results are not clearly presented and Results section should be substantionally improved prior publication.
I would suggest to completely remove Tables from the paper and present all the obtained results using box-plots. I would also suggest to put results for all groups in one figure (not only for one group as in Figure 2).
I would also suggest to use abbreviations for each treatment group. These abbreviations should be introduced in M&M (and Figure 1) and should be used consistently through the manuscript (current abbreviations are not introduced in Figure 1)
Author Response
We would like to thank the reviewers for all the useful comments/suggestions regarding our paper. We have followed the suggestions carefully to improve our manuscript.
We hope that the revised version of our paper will satisfy the requirements for publishing in Journal Coatings.
Yours sincerely,
Lavinia Luminita Voina Cosma

Reviewer 2 Report
The problem considered at work is very important and undertaken for many years by scientists. The article deals with the problem of enamel white spot lesions. The problem is current and very significant in light of the present orthodontic treatment. There are many scientific works that try to use different methods and technologies. The authors place their hope in peptide p11-4, bioactive glass toothpaste, and local fluoride used in conjunction with low-level laser therapy (LLLT). Experimental tests, i.e. surface, roughness, mineral content, and esthetic were performed.
The manuscript is interesting. However, there are several points that I would like to address:
Abstract:
units are missing for numerical values
Methods:
there is no information about which teeth were selected for examinations,
on what basis was such a composition of saliva selected and what was the significance of this composition in experimental studies,
what were the dimensions of the surface of the tested teeth,
how the roughness was determined (e.g. distance, surface area…) - data is missing in this range,
what was the initial surface Ra value,
there is no data on the chemical composition of the preparations used, which is the main issue/basis for discussing the obtained test results,
How many repetitions were made during TGA, XRD, and FTIR tests?
Results:
Units for Ra are missing
there is no explanation of abbreviations used in all tables, e.g. Ra_i, Ra_f in table 1.
Conclusions
how the authors will justify the title of the paper in the context of the results presented in the paper, especially in the context of the formulation of “novel technology”. The paper does not explain what should be understood by this concept.
References
incomplete bibliographic data, e.g. item 9
Author Response

(The authors gave the same response as above.)

Reviewer 3 Report
1. In non-published materials for this publication, please use all the English languages. Especially on its first page.
2. Uppercase and lowercase on the title need to be corrected according to the MDPI format.
3. Keywords need to be reordered based on alphabetical order.
4. What is the novel of the present work? In vitro studies of enamel remineralization have been widely studied in the past. The authors need to highlight their novelty more advance in the introduction section.
5. The authors need to explain the previous work with their findings, novelty, and shortcoming to show the state of the art of the present study.
6. In the introduction section, the authors need to explain in vitro, in vivo, and in silico in at least one paragraph. It should include an explanation of the advantage and disadvantages of the three of them.
7. Why present study only performed in vitro study? It needs more explanation in the introduction section.
8. The authors need to enrich their explanation to describe the potential of using in silico-based study. This important point needs to deliver. Also, to support this explanation, the suggested reference published by MDPI should be adopted as follows: In Silico Contact Pressure of Metal-on-Metal Total Hip Implant with Different Materials Subjected to Gait Loading. Metals (Basel). 2022, 12, 1241. https://doi.org/10.3390/met12081241
9. The objective of the present study needs to be explained in one separate paragraph
10. The authors use the term “Novel technology”, but the authors did not explain the novel that something really new in the existence of the present study. The introduction is too short that only 1 paragraph. With too low-quality introduction, the terms “Novel technology” is something flaws. Recommended using 6 (six) or more paragraphs.
Author Response

(The authors gave the same response as above.)

Round 2
Reviewer 1 Report
The authors have corrected the manuscript according to the suggestions and the paper is now suitable for publication.
Author Response
Thank you very much for taking the time to assess our manuscript and for all the useful comments/suggestions. We have addressed all the concerns that the reviewers raised.
We hope that the revised version of our paper will satisfy the requirements for publishing in Journal Coatings
Yours sincerely,
Lavinia Luminita Voina Cosma

Reviewer 2 Report
The manuscript was improved and corrected almost due to all my comments.
Only one thing must be improved: line 121-122: "was the tested surface of the tooth and had a size of approximately 8x10mm². The note 8x10mm² is incorrect. It might be 8x10 mm (sample dimensions) or 8-10 mm² (sample surface). I hope the Authors understand the difference in the notation.
Author Response

(The authors gave the same response as above.)

Reviewer 3 Report
Good job to The authors, but it has some issue in the revised form that should be addressed.
1. The workflow of the present study needs to be explained in form of a figure to better explain rather than only a dominant paragraph.
2. The result of the present study needs to be compared and discussed with the results from similar previous published literature.
3. Tools used in the present study need to explain the data related to manufacturer, location, country, etc.
4. More information regarding tools' accuracy and tolerance should be given.
5. In the whole of the manuscript, the authors sometimes made a paragraph only consisting of one or two sentences that made the explanation not clearly understood. The authors need to extend their explanation to become a more comprehensive paragraph. In one paragraph, it is recommended to consist of at least 3 sentences with 1 sentence as the main sentence and the other sentences as supporting sentences. For example, line 101-104. Novel description should be extend for clearly understanding.
6. For extending content related in line 78-100, The Reviewer encourage to explain computational study (in silico) would be the basis/preliminary study for further research such as in in vitro and/or in vivo since it is can examine various aspects and parameters in a relatively long time without large costs so as to minimize the effort for trial and error from in vitro and in vivo research. The authors has explain in vitro, in vivo, and in silico, but not utilization in silico as preliminary study. The introduction and/or discussion part of an article should contain this crucial information. In addition, to support this explanation, the MDPI-suggested reference should be included as follows: Computational Contact Pressure Prediction of CoCrMo, SS 316L and Ti6Al4V Femoral Head against UHMWPE Acetabular Cup under Gait Cycle. J. Funct. Biomater. 2022, 13, 64. https://doi.org/10.3390/jfb13020064
7. The limitation of the present study is too short. More explanation is needed.
8. The conclusion needs more elaboration since it is not captured the whole of the present study.
9. A take-home message needs to be added in the abstract and conclusion section.
10. Further research needs to state in the conclusion section.
11. Please used the MDPI format properly, one example the table presentation is not appropriate, and some that have been mentioned before.
12. English needs to be proofread due to grammatical issue and English style. MDPI English editing service would be a solution.
13. The reference needs to be enriched from the literature published in lasth five years. Adopted the reference from MDPI is strongly encouraged.
Author Response

(The authors gave the same response as above.)

Round 3
Reviewer 3 Report
Good job to the Authors.